

# The role of methane in projections of 21st century stratospheric water vapour

Laura Revell[1,2], Andrea Stenke[2], Eugene Rozanov[2,3], William Ball[2,3], Stefan Lossow[4], and Thomas Peter[2]

[1]Bodeker Scientific, Christchurch, New Zealand
[2]Institute for Atmospheric and Climate Science, ETH Zurich, Zurich, Switzerland
[3]Physical-Meteorological Observatory/World Radiation Center, Davos, Switzerland
[4]Karlsruhe Institute of Technology, Karlsruhe, Germany

*Correspondence to:* Laura Revell (laura@bodekerscientific.com)

**Abstract.** Stratospheric water vapour (SWV) is an important component of the Earth's atmosphere as it affects both radiative balance and the chemistry of the atmosphere. Key processes driving changes in SWV through the 21st century include dehydration of air masses transiting the cold-point tropopause (CPT) and methane oxidation. Increasing surface temperatures may strengthen the Brewer-Dobson circulation, such that more methane is transported into the stratosphere where it can be oxidised
to SWV. We use a chemistry-climate model to simulate changes in SWV through the 21st century following the four canonical Representative Concentration Pathways (RCPs). Furthermore, we quantify the contribution that methane oxidation makes to SWV following each of the RCPs. The methane contribution to SWV maximises in the upper stratosphere, however modelled SWV trends are found to be driven predominantly by warming of the CPT and strengthening of the Brewer-Dobson circulation rather than by increasing methane oxidation. SWV changes by -5% to 60% (depending on the location in the atmosphere and
emissions scenario) and increases in the lower stratosphere in all RCPs through the 21st century. Because the lower stratosphere is where water vapour radiative forcing maximises, SWV's influence on surface climate is also expected to increase through the 21st century.

## 1  Introduction

Stratospheric water vapour (SWV) plays an important role in the Earth's radiative balance (Hartmann et al., 2013) such that
decreases in SWV concentrations during the 2000s may have slowed surface temperature increases over the same period (Solomon et al., 2010). Increases in SWV lead to stratospheric ozone depletion both by enhancing ozone-depleting odd hydrogen cycles, and by increasing the prevalence of polar stratospheric clouds (PSCs) which facilitate polar springtime ozone depletion (Kirk-Davidoff et al., 1999; MacKenzie et al., 2004; Stenke and Grewe, 2005). SWV owes its existence primarily to transport from the troposphere, which occurs predominantly through the tropical cold-point tropopause (CPT) (Brewer, 1949;
Holton and Gettelman, 2001), and to methane oxidation via Reaction R1 (Bates and Nicolet, 1950; Le Texier et al., 1988):

$$CH_4 + OH \rightarrow CH_3 + H_2O \tag{R1}$$



Methane oxidation is an important in-situ source of water vapour in the middle and upper stratosphere (Brasseur and Solomon, 2005). Further oxidation of the methyl radical $CH_3$ leads to some additional $H_2O$ formation (although this is limited by the formation of other H-containing molecules, such as $H_2$ or OH). Between 1980 and 2010, balloon-borne measurements of SWV over Boulder, Colorado (40°N) showed an increase in SWV of ∼1 ppmv (almost 30%) between 16-26 km, with ∼0.25 ppmv of the net increase due to increased methane oxidation in the stratosphere (Hurst et al., 2011). This inferred contribution of methane oxidation to the SWV trend is in good agreement with an earlier analysis of balloon-borne SWV measurements from Japan, France and Sweden (launch sites were between 39-68°N), which concluded that methane contributed 25-34% to the increase in water vapour in the middle stratosphere between 1978 and 2003, and a smaller amount (1.6-10.7%) in the lower stratosphere (Rohs et al., 2006). The remaining increase was attributed to increased transport of water vapour from the troposphere to the stratosphere. More recently, Hegglin et al. (2014) showed that merged satellite SWV observations display positive trends globally in the upper stratosphere due to methane increases and changes in stratospheric circulation, but negative trends in the lower stratosphere.

Austin et al. (2007) also investigated the causes of past changes in SWV, but used a coupled chemistry-climate model (CCM) to do so. In contrast to the measurement-based studies of Hurst et al. (2011) and Rohs et al. (2006), they found that methane oxidation was primarily responsible for elevated SWV concentrations between 1960 and 2000. However, their model exhibited a tropical tropopause cold bias of 3-4 K (Eyring et al., 2006), and SWV concentrations in the lower stratosphere were about 30% lower than expected from observations, implying that too little water vapour entered the stratosphere through the tropical CPT.

The minimum temperature experienced along air parcel trajectories transiting the tropical tropopause into the stratosphere (the CPT) exerts a strong influence on water vapour concentrations in the lower stratosphere (Randel et al., 2004; Fueglistaler et al., 2005; Garcia et al., 2007). Through the 21[st] century, the CPT is projected to warm (Gettelman et al., 2010), and therefore water vapour concentrations are expected to increase in the lower stratosphere. Transport and evaporation of ice through the tropical tropopause is also a significant process for water vapour entering the stratosphere (Dessler et al., 2016). Because SWV is a greenhouse gas, increasing lower stratospheric humidity is anticipated to lead to further warming of the troposphere; Dessler et al. (2013) estimate that the climate sensitivity of this feedback is $0.3\,Wm^{-2}K^{-1}$, with approximately $0.1\,Wm^{-2}K^{-1}$ coming from increases in SWV through the CPT and the rest coming from increases in water vapour entering the lowermost stratosphere through the extratropical tropopause.

Climate models project that increasing surface temperatures will increase the rate of tropical upwelling (Butchart and Scaife, 2001; Butchart et al., 2010), so that more air is transported into the cold upper tropical troposphere with a significantly extended methane lifetime and subsequently more methane capable of entering the stratosphere; this will further affect SWV (Austin et al., 2007). However, Randel et al. (2006) and Dhomse et al. (2008) showed that enhanced tropical upwelling can lead to smaller concentrations of ozone in the tropical lower stratosphere, and therefore colder temperatures and lesser amounts of stratospheric water vapour.

SWV will also be affected by methane emissions through the 21[st] century. Since 1750 until the present day, atmospheric methane concentrations have increased by a factor of ∼2.5 (Montzka et al., 2011). The Representative Concentration Pathways



(RCPs) used by the Intergovernmental Panel on Climate Change (IPCC) follow a range of methane concentration scenarios through the 21$^{st}$ century, depending on the assumptions made regarding future emissions mitigation (van Vuuren et al., 2011a). While under the RCP 2.6 scenario methane reduces by one-third below its year 2000 value by 2100, concentrations more than double under RCP 8.5 over the same period (see Fig. 8a, later).

Oman et al. (2008) studied future changes in SWV using a CCM, and showed that increasing methane concentrations under the IPCC's SRES A1B scenario for greenhouse gases (Nakicenovic and Swart, 2000) drove the majority of the simulated increase in SWV until the mid-21$^{st}$ century. The methane effect had a greater effect on SWV increases compared with direct water vapour injections following a warming of the tropical tropopause or a widening of the tropical upwelling (Rosenlof, 2002). Although methane concentrations under the A1b scenario decrease through the second half of the 21$^{st}$ century, expected

decreases in SWV were balanced by increased tropical upwelling, leading to a near-zero trend.

       Clearly future SWV trends are closely linked to future methane emissions, however little attention has been given to the effects that a range of methane scenarios may have on SWV evolution. Here we use CCM simulations to first isolate methane's chemical contribution to SWV from climate effects, then show projections of SWV following the four RCPs in the 21$^{st}$ century, and methane's contribution to these projections.

**2    Computational Methods**

Model simulations were performed with version 3 of the SOCOL (SOlar Climate Ozone Links) CCM (Stenke et al., 2013; Revell et al., 2015). SOCOL v.3 consists of MA-ECHAM5, the middle atmosphere version of the ECHAM general circulation model (Roeckner et al., 2003), and the MEZON chemistry model (Egorova et al., 2003). For the current study, the model was run as described by Revell et al. (2015), i.e. with T42 horizontal resolution (grid cell sizes correspond to approximately 2.8° ×

2.8°) and 39 vertical levels between Earth's surface and 0.01 hPa (~80 km).

       Cloud and ice formation and sedimentation is handled by the MA-ECHAM5 cloud scheme as described by Roeckner et al. (2003), except for in the polar lower stratosphere (that is, above the tropopause pole-wards of 50°). Here, ice formation and sedimentation is handled by SOCOL v.3's PSC scheme, which is necessary to avoid 'normal' cirrus formation in PSC regions by the MA-ECHAM5 cloud scheme, and to ensure consistent treatment of PSCs in the model. This is a different approach

to that used in SOCOL v.2 (Schraner et al., 2008), which participated in CCMVal-2 (SPARC CCMVal-2, 2010). SOCOL v.3 produces a more realistic simulation of water vapour than v.2, as shown by Stenke et al. (2013).

       The model simulations presented here (with the exception of the RCP 6.0-fEmis-fCH$_4$ simulation; see Table 1) were performed in support of phase one of the IGAC/SPARC Chemistry-Climate Model Initiative (CCMI-1) (Eyring et al., 2013; CCMI, 2015). Boundary conditions conform to CCMI-1 requirements, and are summarised in Table 1. All simulations used

the World Meteorological Organization's (WMO) A1 scenario for ozone-depleting substances (World Meteorological Organization, 2011). Greenhouse gases (including carbon dioxide, nitrous oxide, methane and halocarbons) were prescribed as surface concentrations, while ozone precursor gases (including nitrogen oxides, carbon monoxide and non-methane volatile organic compounds) were prescribed as emission fluxes. Sea surface temperatures were prescribed following output from the



CESM1(CAM5) earth system model (Meehl et al., 2013). Year 2000 stratospheric aerosol surface area densities and optical parameters were prescribed from the SAGE_4$\lambda$ dataset (Arfeuille et al., 2013; Luo, 2013) for each year of all of the simulations, as the year 2000 was a volcanically quiescent time.

## 3   Results and Discussion

### 3.1   Comparison with observations

We first of all assess the fidelity of SOCOL v.3's simulation of water vapour, temperature and methane through comparison with observations. Although Stenke et al. (2013) rigorously evaluated SOCOL v.3 in this respect, the model formulation was updated for CCMI leading to some important differences in the output. Notably, the update of reaction rate coefficients to the most recent recommendations by Sander et al. (2011) leads to suppressed $HO_x$ (H+OH+$HO_2$) formation, and, therefore, increased stratospheric ozone and extra warming in the upper stratosphere and mesosphere. Overall the extra warming in the current model version leads to better agreement between modelled and observed upper atmospheric temperatures than previous model formulations (not shown). In addition, the parameterisation of shortwave heating by $O_2$ and $O_3$ absorption in the mesosphere applied in Stenke et al. (2013) was updated and extended according to Sukhodolov et al. (2014), which also contributes to warming in the mesosphere.

Figure 1 examines dehydration processes near the tropical tropopause by comparing the modelled annual cycle in equatorial temperature and water vapour at the CPT with observations. In SOCOL v.3 the WMO-defined tropopause (the lowest level at which the lapse rate decreases to 2 °C km$^{-1}$ or less) typically sits at 100 hPa in the tropics, while the CPT typically locates at a lower pressure of 90 hPa. Figure 1a compares modelled CPT temperatures with radiosonde measurements (Seidel et al., 2001). Between January and September the modelled temperatures are 1-2 K colder than observations, although agree within one standard deviation, and are in closer agreement for the remainder of the year. Modelled water vapour entering the stratosphere agrees with MIPAS observations (Schieferdecker et al., 2015) within one standard deviation (Fig. 1b), although the modelled annual cycle is shifted such that the annual maximum occurs a month earlier, in September rather than October (discussed further below). Another possible reason for the phase shift may be the lower vertical resolution of the satellite data set.

Figure 2 compares modelled water vapour with MIPAS observations higher in the atmosphere. Modelled water vapour profiles generally show good agreement with observations (Fig. 2a-c), although the modelled equatorial stratosphere at 30 hPa is up to 20% (0.78 ppmv) too moist, depending on the season (Fig. 2b). As demonstrated in Fig. 1, a too-moist stratosphere cannot be due to excess water vapour entering the stratosphere via the CPT as the modelled CPT is cold-biased. Furthermore the annual cycle in the lower to middle stratosphere is out-of-phase compared with observations (Fig. 2d). The annual cycle in water vapour progresses with decreasing pressure (Randel et al., 1998), and because SOCOL v.3 has too-fast upward propagation (estimated from the water vapour tape recorder (Stenke et al., 2013)), seasonal variability is shifted upwards such that maximum water vapour mixing ratios occur out-of-phase with observations. However when examining the annual mean, SOCOL v.3 compares favorably with observations (Fig. 2e). Because the aim of this study is to analyse future changes in SWV on a multi-decadal scale, we do not anticipate that the model's shifted annual cycle in SWV will substantially bias our results.



In the upper stratosphere, SWV is produced as a result of methane oxidation. SOCOL v.3 is in good agreement with observations within one standard deviation throughout the stratosphere (Fig. 3), except above 2 hPa in the equatorial atmosphere (Fig. 3b) where it exhibits a positive bias in methane of ∼50% compared to HALOE observations (Grooß and Russell, 2005). Recently, Laeng et al. (2015) and Plieninger et al. (2016) have shown that HALOE may be low biased compared to other

satellite methane observations in the upper stratosphere and lower mesosphere. In this region of the atmosphere, SOCOL v.3 agrees favourably with MIPAS observations (not shown).

### 3.2   Drivers of SWV change

Drivers of SWV change are shown in Figure 4 for the RCP 6.0-fEmis and RCP 6.0-fEmis-fCH$_4$ simulations. These simulations are both based on RCP 6.0, although tropospheric ozone precursor emissions (including nitrogen oxides, non-methane volatile

organic compounds and carbon monoxide) are held constant at 1960 levels (Table 1). Therefore in the stratosphere RCP 6.0-fEmis is very similar to RCP 6.0 (at least in terms of SWV). The only difference between the RCP 6.0-fEmis and RCP 6.0-fEmis-fCH$_4$ simulations is that methane is kept at constant 1960 concentrations throughout the RCP 6.0-fEmis-fCH$_4$ simulation, while in the RCP 6.0-fEmis simulation methane follows RCP 6.0 (Fig. 4a). Comparing RCP 6.0-fEmis and RCP 6.0-fEmis-fCH$_4$ therefore isolates the chemical impact of methane on SWV.

Figure 4b shows that the rate of tropical upwelling is projected to increase through the 21$^{st}$ century, as simulated by most CCMs (Butchart et al., 2010; SPARC CCMVal-2, 2010). Stronger tropical upwelling transports more methane from the troposphere into the stratosphere where it undergoes subsequent oxidation to water vapour via Reaction R1 (Austin et al., 2007). Stronger tropical upwelling has also been shown to lead to reduced ozone concentrations in the tropical lower stratosphere (Bekki et al., 2013), resulting in cooling of the CPT and subsequent decreases in SWV (Randel et al., 2006; Dhomse et al.,

2008). Although tropical lower stratospheric ozone decreases over time in our model simulations (not shown), we see increases in both annual-mean temperatures at the CPT (Figure 4c) and water vapour concentrations just above the tropical CPT (Figure 4d). Between 1960 and 2100, annual-mean CPT temperatures increase by approximately 1.35 K in the RCP 6.0-fEmis and RCP 6.0-fEmis-fCH$_4$ simulations (Fig. 4c), resulting in an increase in stratospheric entry water vapour of approximately 1.05 ppmv (∼30%) over the same period. These changes are similar to those simulated by the CCMVal-2 models, which projected

increases in the CPT temperature and stratospheric entry water vapour of ∼1.4 K and 0.7-1.4 ppmv, respectively, between 1960 and 2100 (Gettelman et al., 2010), following the IPCC's SRES A1B scenario for greenhouse gases (which is similar to RCP 6.0 in terms of greenhouse gas concentrations) (Nakicenovic and Swart, 2000).

### 3.3   Impacts of methane and climate change on SWV

Figure 5a shows the change in SWV between the 1960s and 2090s decades of the RCP 6.0-fEmis-fCH$_4$ simulation. SWV

increases by ∼1 ppmv (or 20%) through much of the stratosphere with the exception of the Antarctic lower stratosphere. Here, there is no change in SWV because this region is near saturation during winter, and so as the stratosphere cools through the 21$^{st}$ century more irreversible loss of SWV occurs thus cancelling out increases in water vapour entering the stratosphere (Oman et al., 2008; Dessler et al., 2013). The ∼1 ppmv change in SWV seen between 1960 and 2100 is fairly uniform throughout



most of the stratosphere as transport within the stratosphere leads to homogenisation at a faster rate than the rate at which entry water vapour changes (Dessler et al., 2013).

As methane concentrations were kept constant in the RCP 6.0-fEmis-fCH₄ simulation, the contribution of methane oxidation to changes in SWV is excluded. Potential drivers of the SWV changes shown in Fig. 5a therefore result from some combina-
tion of increasing CPT temperatures allowing more water vapour to enter the stratosphere, the strengthening Brewer-Dobson circulation transporting more methane into the stratosphere where it undergoes oxidation to SWV, and temperature-induced changes in the rate of methane oxidation. Between 1960 and 2100, stratospheric temperatures are projected to cool by up to 12 K in the RCP 6.0-fEmis-fCH₄ simulation, with the most cooling projected for the upper stratosphere (Fig. 5c). As a result, the rate constant for Reaction R1 is projected to slow by up to 3.5% in the upper stratosphere (not shown). We therefore do not
expect temperature-induced changes in the rate of methane oxidation to significantly influence future SWV changes.

Dessler et al. (2013) showed the change in SWV over the 21$^{st}$ century with the contribution from methane oxidation removed (by assuming that each methane molecule destroyed produces two water molecules). This is akin to looking at the change in SWV in our RCP 6.0-fEmis-fCH₄ simulation (Fig. 5a) and indeed the results are very similar, with both the GEOSCCM (Goddard Earth Observing System Chemistry Climate Model Dessler et al., 2013) and SOCOL v.3 models simulating an
increase in SWV due to climate-related changes of ∼1 ppmv through the 21$^{st}$ century.

Fig. 5b shows the difference in SWV in the 2090s between the RCP 6.0-fEmis and RCP 6.0-fEmis-fCH₄ simulations. The methane increase that occurs between 1960 and 2100 in the RCP 6.0-fEmis simulation is responsible for statistically significant increases in SWV throughout the upper (and much of the middle) stratosphere, maximising at approximately 0.8 ppmv (∼12%) in the extra-tropical upper stratosphere. Note that in RCP 6.0, the maximum methane concentrations (∼1.97
ppmv) occur around 2070, and by 2100 methane is projected to have decreased to 1980 levels. The contribution of methane to SWV is small and statistically insignificant in the lower stratosphere (noted also by Rohs et al. (2006)), which is where SWV exerts the greatest impact on radiative forcing (Solomon et al., 2010).

To summarise, climate-related changes are responsible for an increase in SWV of 1 ppmv between 1960 and 2100. Changes in methane are responsible for an increase of almost a similar magnitude (0.8 ppmv), but only in the upper stratosphere;
closer to the tropopause methane does not induce such a large increase in SWV, as found in previous CCM studies (Tian and Chipperfield, 2006; Oman et al., 2008).

### 3.4 Projections for the RCPs

While the simulations discussed so far (RCP 6.0-fEmis and RCP 6.0-fEmis-fCH₄) are both based on RCP 6.0, methane follows different concentration pathways in the other RCPs (Fig. 8a). To isolate the effect of methane in the other RCPs, we
approximated SWV in a way similar to Austin et al. (2007) and Oman et al. (2008):

$$H_2O(\theta, p, t) = H_2O|_e(t - \tau) + 2[CH_4|_0(t - \tau) - CH_4(\theta, p, t)] \tag{1}$$

where for each latitude ($\theta$), pressure level ($p$) and time ($t$), $H_2O|_e$ is the tropical (10°N-10°S) water vapour concentration at 70 hPa, $\tau$ is the age of air and $CH_4|_0$ is the tropical methane concentration at 150 hPa. The first term in Eq.(1) (i.e., $H_2O|_e$ ($t$ -



$\tau$)) represents water vapour concentrations entering the stratosphere, and the second term represents the contribution to SWV from methane oxidation. The age of air for each latitude, pressure and time ($\tau(\theta,p,t)$) is included because for a given parcel of air, entry into the stratosphere will have occurred at an earlier time (depending on where the parcel of air is), and the amount of methane oxidation that will have occurred depends on the time since that parcel of air resided in the troposphere (Austin et al., 2007).

To demonstrate the validity of the approximation underlying Eq.(1), SWV constructed using Eq.(1) is compared with actual SWV for the RCP 6.0-fEmis simulation, for a range of latitudes and pressure levels (Fig. 6). Constructed SWV shows larger variability than actual SWV because it is constructed using entry water vapour, which exhibits larger modelled variability compared with water vapour higher in the stratosphere. The constructed methane contribution (second term in Eq.(1)) is also shown, compared with the actual methane contribution. Because the constructed methane contribution is calculated by subtracting the stratospheric methane concentration (at any location in the stratosphere) from the tropical methane concentration at 150 hPa, and because modelled stratospheric methane concentrations become more variable higher in the atmosphere (recalling that methane is prescribed as a surface concentration uniformly over the globe), the constructed methane contributions in Fig. 6 exhibit greater variability than the actual methane contribution. The actual methane contribution is obtained by calculating the difference between SWV in the RCP 6.0-fEmis and RCP 6.0-fEmis-fCH$_4$ simulations for each month, and adding the mean methane contribution for the 1960s; noting that methane oxidation makes a small but not insignificant contribution to SWV in the RCP 6.0-fEmis-fCH$_4$ simulation, albeit mostly constant between 1960-2100. Methane contributes to approximately one-third of upper stratospheric water vapour in the RCP 6.0-fEmis simulation, although the contribution is smaller in the lower stratosphere, especially in the tropics. Small decreases in the methane contribution are observed after ~2080 following decreases in surface methane (as prescribed by RCP 6.0).

We now examine SWV in the four RCPs between 2000 and 2100 (Fig. 7). As for Fig. 6, SWV time series constructed with the approximation in Eq.(1) agree well with the actual SWV simulated in the RCPs (not shown). The projections of SWV for RCP 6.0 in Fig. 7, and the RCP 6.0-fEmis simulation in Fig. 6 are very similar as these simulations differ only with respect to tropospheric ozone precursors (Table 1). Large differences in SWV between the RCPs are projected for the end of the 21$^{st}$ century; for example, at 1 hPa SWV is 3-4 ppmv greater in RCP 8.5 compared with RCP 2.6 (depending on latitude) (Fig. 7a-c). Of all the RCPs, the largest increase in SWV through the 21$^{st}$ century at any pressure level and latitude is projected to occur in RCP 8.5. The RCP 8.5 scenario prescribes the largest increase in methane of ~2 ppmv or 114% between 2000-2100 (Fig. 8a), but also the largest increases in the other greenhouse gases (especially CO$_2$), which drive a faster rate of tropical upwelling (Fig. 8b) and warm the CPT. Between 2000-2100 the CPT warms by 0.7 K in RCP 2.6, and 2.4 K in RCP 8.5 (Fig. 8c). Because it warms more, the amount of water vapour entering the stratosphere increases at a larger rate in RCP 8.5 compared with the other RCPs; between the 2000-2100 entry SWV increases by 1.6 ppmv (42%) in RCP 8.5, compared with 0.4 ppmv (12%) in RCP 2.6 (Fig. 8d).

To better understand the contribution that methane makes to SWV concentrations in the RCPs, Fig. 9 shows the methane contribution term (calculated from Eq.(1)), expressed as a percentage of SWV. As previously stated, methane makes the largest contribution to the SWV budget in the upper stratosphere (~40-50% following RCP 8.5, Fig. 9a-c), and the smallest contribu-





tion in the tropical lower stratosphere (∼4% following RCP 2.6, Fig. 9h). In all simulations other than RCP 8.5, the fractional contribution of methane to SWV decreases through the 21$^{st}$ century, despite the fact that faster tropical upwelling transports more methane into the stratosphere in all simulations (Fig. 8b). Primarily this is because prescribed surface methane concentrations decrease after approximately 2015, 2050 and 2080 for RCPs 2.6, 4.5 and 6.0 respectively (Fig. 8a). Secondly, the rate of methane oxidation (Reaction R1) is temperature-dependent and slows as the stratosphere cools through the 21$^{st}$ century, although this is not a significant process as discussed in Sect. 3.3. Finally, under RCP 8.5, more water vapour enters the stratosphere due to warming of the CPT, such that the fractional contribution of methane plateaus. This increase in concentrations of water vapour entering the stratosphere is visible in the tropical lower stratospheric regions of Fig. 10. Indeed, in RCPs 2.6, 4.5 and 6.0, the largest changes in SWV between 2000 and 2100 are due to increasing concentrations of water vapour entering the tropical lower stratosphere, while in RCP 8.5 (Fig. 10d) the largest SWV increase (up to 60%) occurs in the extra-tropical upper stratosphere, and methane is responsible for just over half of that increase (Figs. 9a and c).

The contribution that methane is projected to make to future SWV evolution is dependent on pressure, latitude, and the methane growth scenario. The biggest contributions are seen in the upper stratosphere (Fig. 9). All of the RCPs project increases in water vapour entering the stratosphere due to warming of the CPT (Figs. 8c-d and 10). Therefore, future increases in SWV will contribute to global warming, given SWV's role as a greenhouse gas, in agreement with Dessler et al. (2013).

In all of the RCPs other than RCP 2.6, upper SWV concentrations increase (Fig. 10). We therefore expect enhanced rates of the ozone-destroying HO$_x$ cycles, which are fastest in the upper stratosphere, although this will not impact the overall increase in stratospheric ozone projected through the 21$^{st}$ century (Pawson and Steinbrecht, 2014). We do not see an enhancement in lower SWV concentrations over the Antarctic in the RCPs (Fig. 10) because this region is near saturation during winter. Therefore, changes in ozone here will likely be due to factors other than SWV. Revell et al. (2012) showed that throughout the middle stratosphere, increases in methane and SWV in the RCPs are expected to lead to increased ozone abundances through the 21$^{st}$ century owing in part to SWV-induced cooling of the stratosphere which slows the gas-phase catalytic ozone loss cycles. Such cooling through the 21$^{st}$ century is also expected to affect circulation patterns in the lower stratosphere and troposphere (Maycock et al., 2013).

## 4 Conclusions

We have used the SOCOL v.3 CCM to simulate changes in SWV through the 21$^{st}$ century and attributed these changes to various processes. Concentrations of water vapour entering the tropical lower stratosphere are projected to increase because of warming of the CPT. Between 1960 and 2100 under RCP 6.0, such climate-related changes are projected to lead to increases in water vapour of approximately 1 ppmv throughout the stratosphere, with the exception of the already-saturated Antarctic lower stratosphere. Higher up in the stratosphere, methane oxidation contributes to the SWV burden, although its contribution is highly dependent on latitude, pressure and the methane concentration scenario. In 2100, the largest contribution methane makes to SWV is ∼50% (following RCP 8.5 in the extra-tropical upper stratosphere) and the smallest contribution is ∼4% (following RCP 2.6 in the tropical lower stratosphere). SWV itself increases everywhere following RCPs 4.5, 6.0 and 8.5



(except in the Antarctic lower stratosphere), with the largest increase of ∼60% occurring in RCP 8.5 in the extra-tropical upper stratosphere. In RCP 2.6, which represents a low emissions pathway for future anthropogenic greenhouse gases, SWV decreases by up to 5% in the upper stratosphere between 2000 and 2100, but increases by up to 10% in the tropical lower stratosphere. Given that water vapour exerts the largest influence on radiative forcing in the lower stratosphere compared with

5   higher altitudes, SWV's influence on surface climate is expected to increase through the 21$^{st}$ century following all of the RCPs.

*Acknowledgements.* SOCOL v.3 CCMI-1 data are held at the British Atmospheric Data Centre, see
http://catalogue.ceda.ac.uk/uuid/1005d2c25d14483aa66a5f4a7f50fcf0. LER thanks Greg Bodeker for helpful comments on this manuscript.
ER appreciates support from the Swiss National Science Foundation (SNSF) under Grant CRSII2-147659 (FUPSOL II). WB was funded by
SNSF grants 200021-149182 (SILA) and 200020-163206 (SIMA).



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





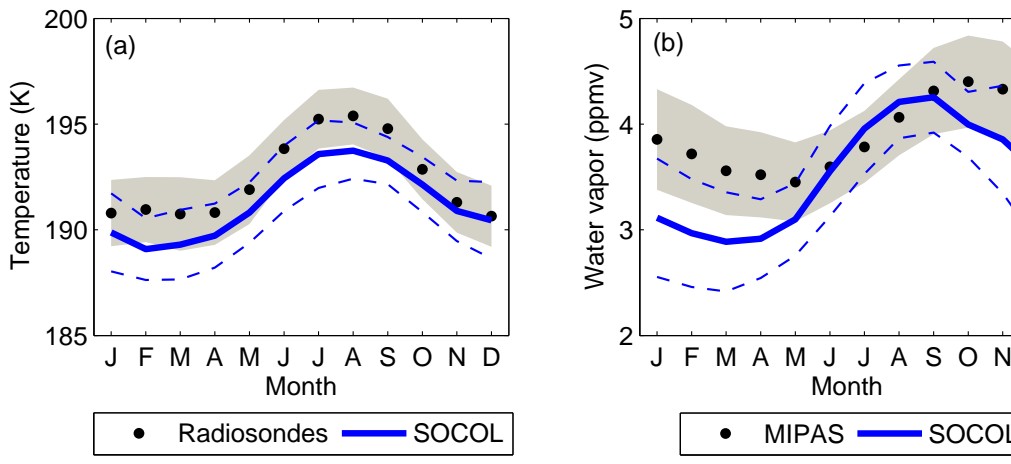

**Figure 1.** Modelled climatological mean annual cycle in equatorial temperature and water vapour from the SOCOL v.3 RCP 6.0 simulation compared with observations. (a) Modelled (4.2°N-4.2°S) zonal-mean CPT temperature at 90 hPa compared with radiosonde measurements of the CPT (5°N-5°S) between 1961-1997. (b) Modelled (4.2°N-4.2°S) zonal-mean water vapour concentrations at 80 hPa compared with MIPAS measurements (7.5°N-7.5°S) between 2005-2012. The grey shaded areas represent one standard deviation either side of the observations and the dashed blue lines indicate one standard deviation either side of model data.

van Vuuren, D. P., Stehfest, E., den Elzen, M. G. J., Kram, T., van Vliet, J., Deetman, S., Isaac, M., Klein Goldewijk, K., Hof, A., Mendoza Beltran, A., Oostenrijk, R., and van Ruijven, B: RCP2.6: exploring the possibility to keep global mean temperature increase below 2°C. Clim. Change, 109, 95-116, doi:10.1007/s10584-011-0152-3, 2011b.

World Meteorological Organization: Scientific Assessment of Ozone Depletion: 2010, WMO Global Ozone Research and Monitoring Project
5    – Report No. 52, Geneva, Switzerland, 2011.



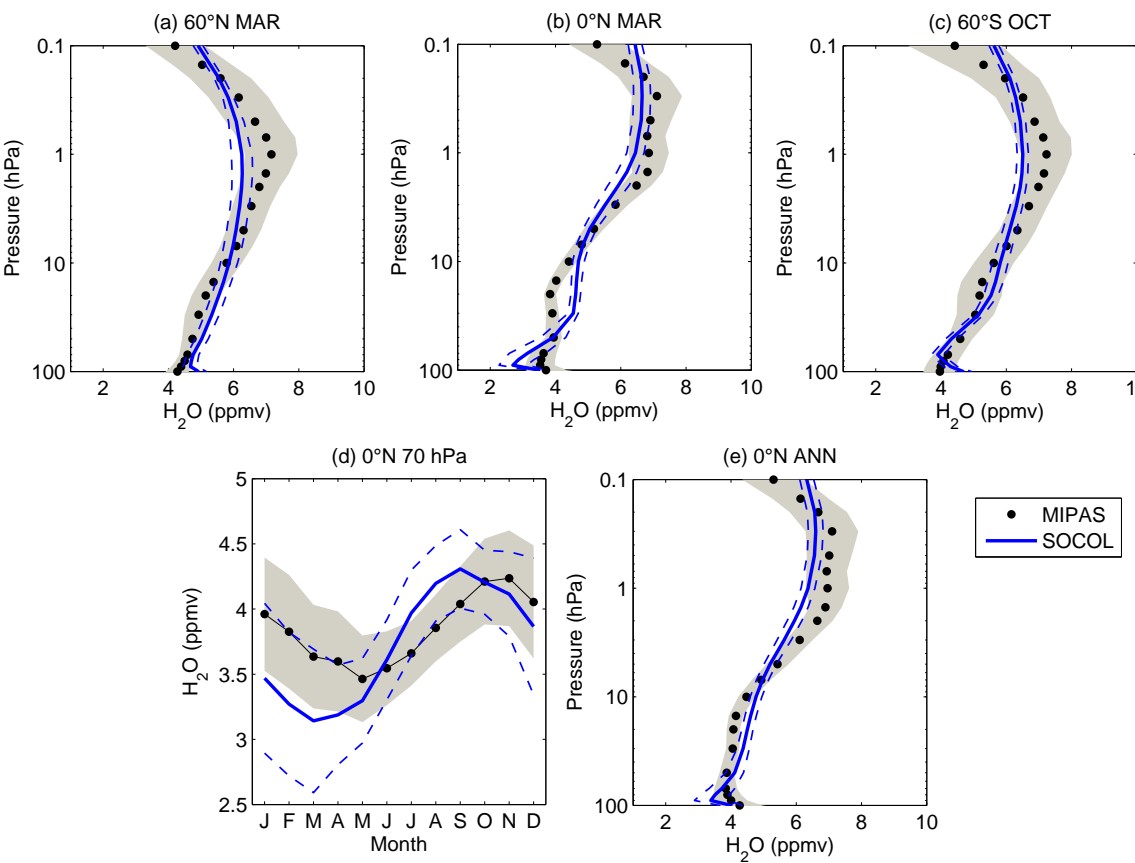

**Figure 2.** Modelled water vapour from the SOCOL v.3 RCP 6.0 simulation compared with MIPAS observations (climatological means for 2005-2012) for (a) 60°N, March; (b) the equator, March; (c) 60°S, October; (d) the equator, 70 hPa; (e) the equator, annual mean. The grey shaded areas represent one standard deviation either side of observations and the dashed blue lines represent one standard deviation either side of model data.



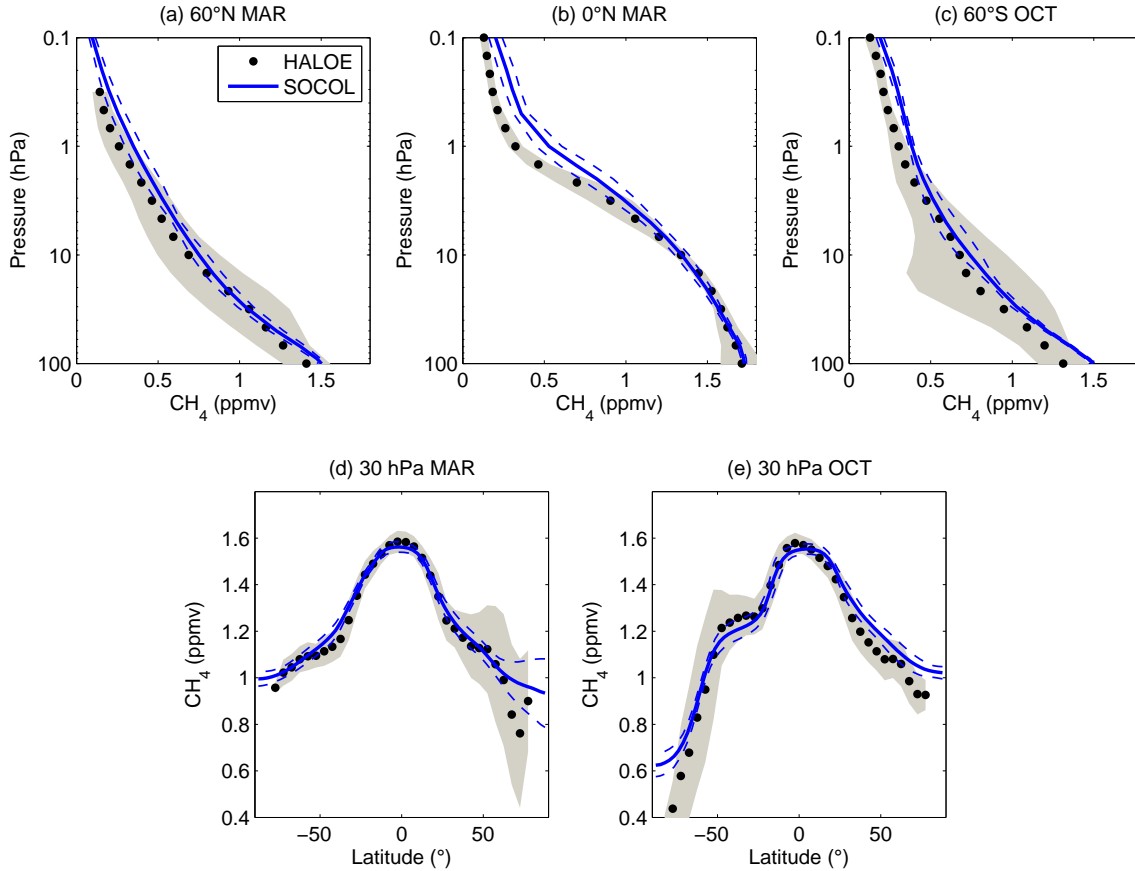

**Figure 3.** Modelled methane concentrations from the SOCOL v.3 RCP 6.0 simulation compared with HALOE observations (climatological means for 1991-2002) for (a) 60°N, March; (b) the equator, March; (c) 60°S, October; (d) 30 hPa, March; (e) 30 hPa, October. The grey shaded areas represent one standard deviation either side of observations and the dashed blue lines represent one standard deviation either side of model data.

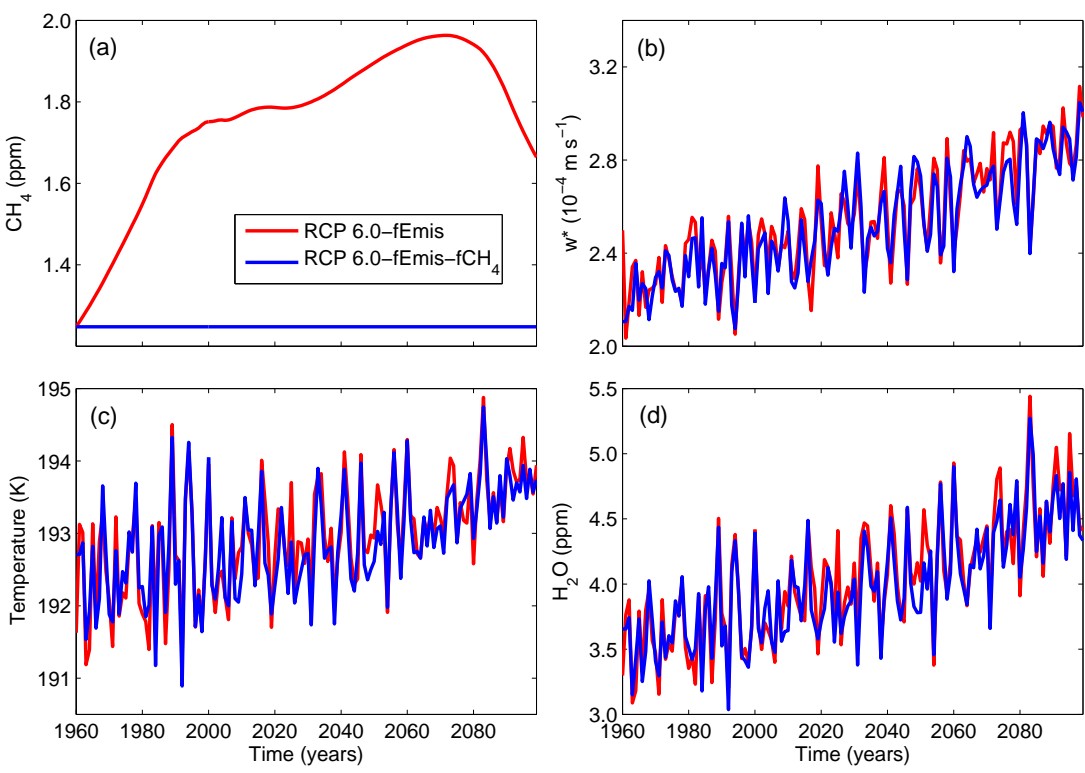

**Figure 4.** Drivers of SWV change in the RCP 6.0-fEmis and RCP 6.0-fEmis-fCH$_4$ simulations. (a) Global-, annual-mean surface methane concentrations; (b) Annual-mean rate of the vertical residual circulation at 70 hPa, 20°N-20°S; (c) Annual-mean tropical CPT temperatures, 20°N-20°S, 90 hPa. (d) Annual-mean stratospheric entry water vapour at 80 hPa, 20°N-20°S.





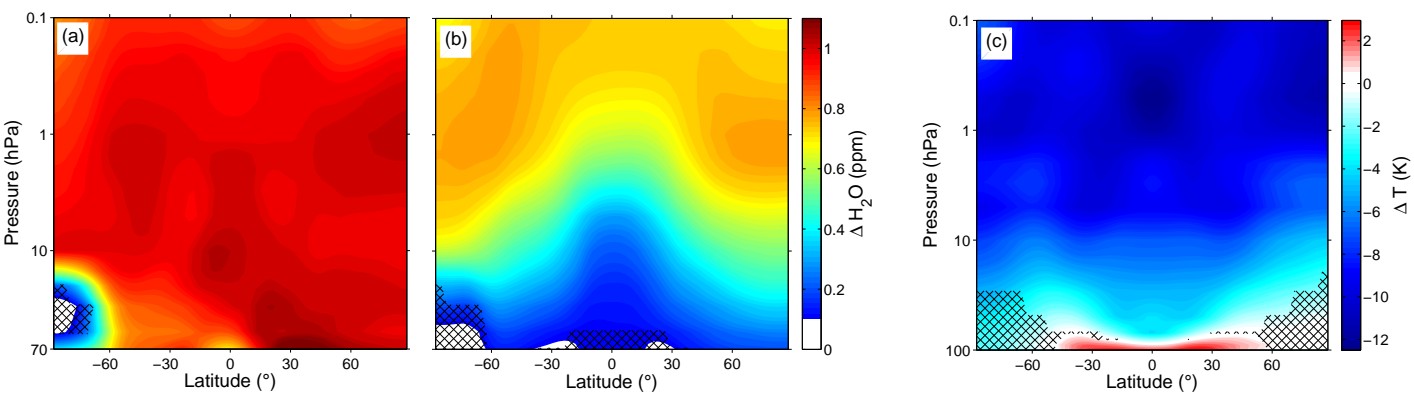

**Figure 5.** Impact of changes in climate and methane on SWV. (a) Increase in water vapour between the 1960s and 2090s in the RCP 6.0-fEmis-fCH$_4$ simulation (2090s minus 1960s); (b) increase in water vapour in the 2090s decade between the RCP 6.0-fEmis and RCP 6.0-fEmis-fCH$_4$ simulations (RCP 6.0-fEmis minus RCP 6.0-fEmis-fCH$_4$); (c) zonal-mean temperature change in the RCP 6.0-fEmis-fCH$_4$ simulation (2090s minus 1960s). Hatching shows where the difference is not statistically significant at the 95% level of confidence, calculated with the Student's t-test.




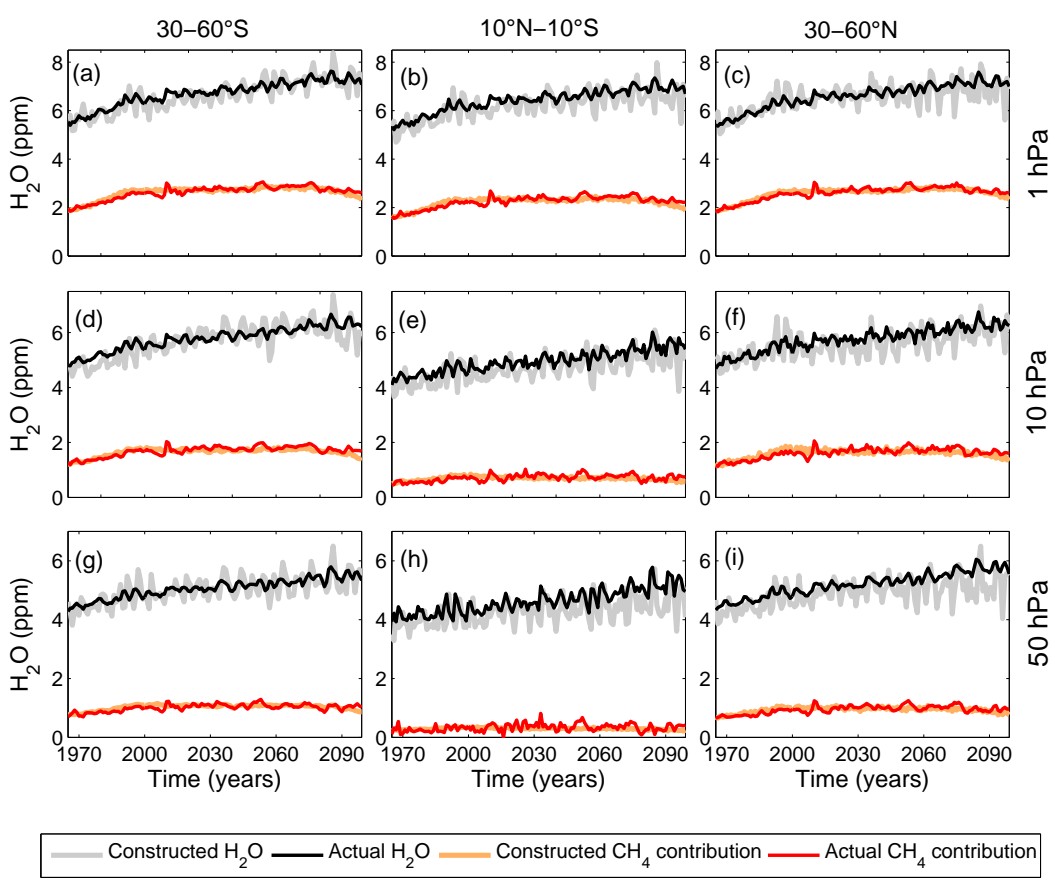

**Figure 6.** Constructed SWV (grey lines), actual SWV (black), the constructed methane contribution (orange) and the actual methane contribution (red) to SWV in the RCP 6.0-fEmis simulation. Top row: upper stratosphere (1 hPa). Center row: middle stratosphere (10 hPa). Bottom row: lower stratosphere (50 hPa). Left column: southern midlatitudes (30-60°S). Center column: tropics (10°N-10°S). Right column: northern midlatitudes (30-60°N). See text for details on how the methane contributions were calculated. All quantities shown are annual means.





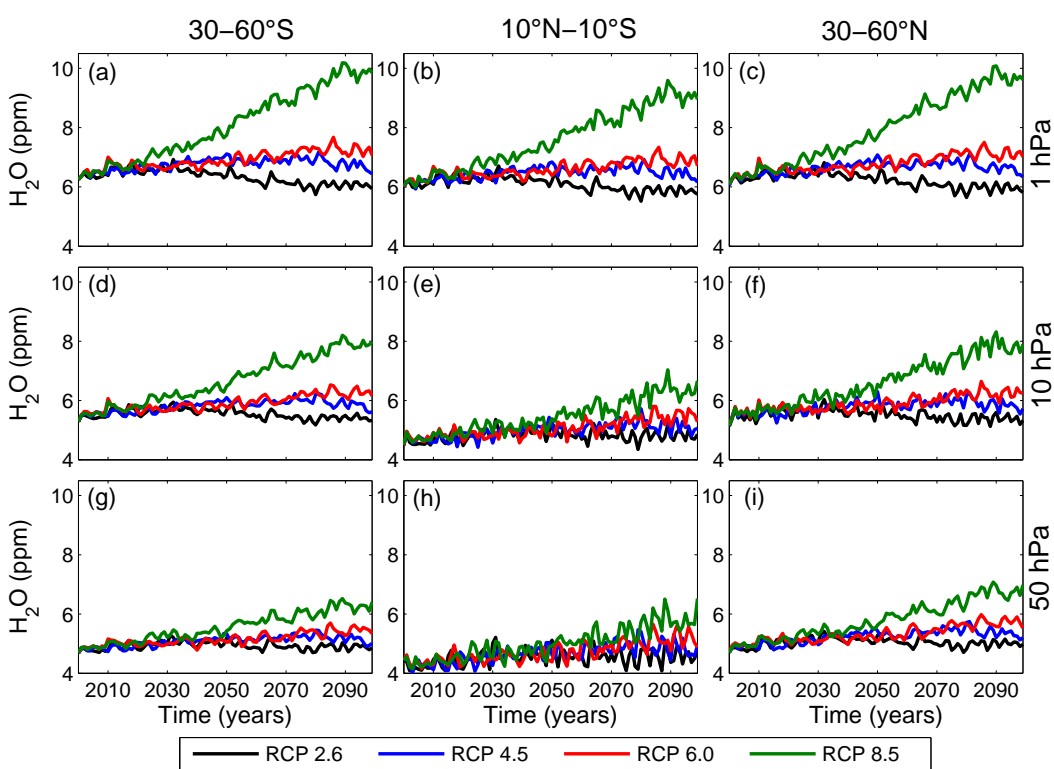

**Figure 7.** SWV in the four RCP simulations for the same regions shown in Fig. 6.





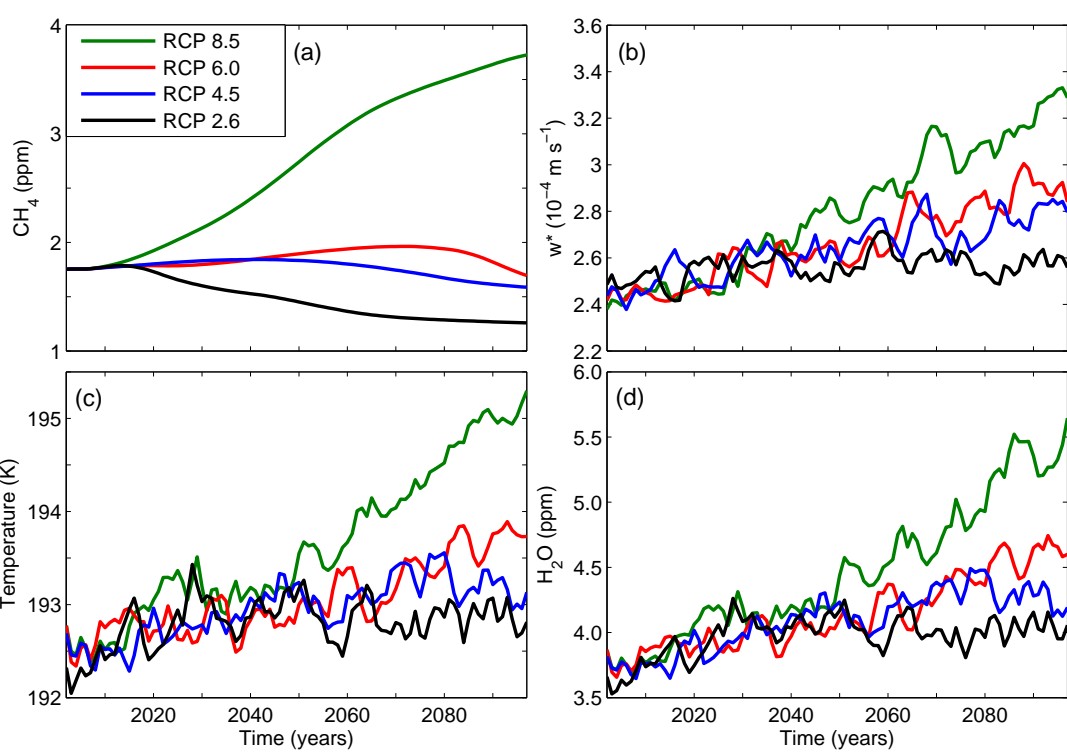

**Figure 8.** Same as Fig. 4, but showing drivers of SWV change in the RCPs. For clarity, the time series in (b)-(d) have been smoothed with a five-year running mean.





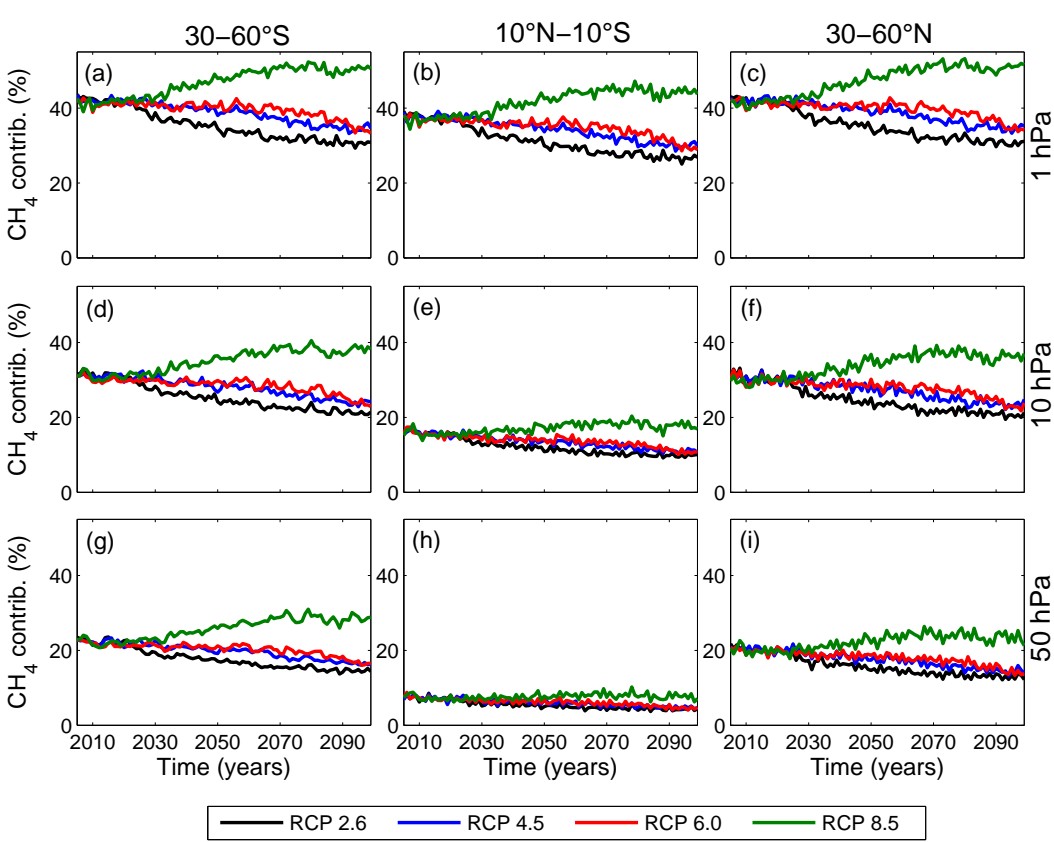

**Figure 9.** Percentage contribution of methane to SWV in the four RCP simulations for the regions shown in Fig. 6. Methane contributions were calculated as for Fig. 6.

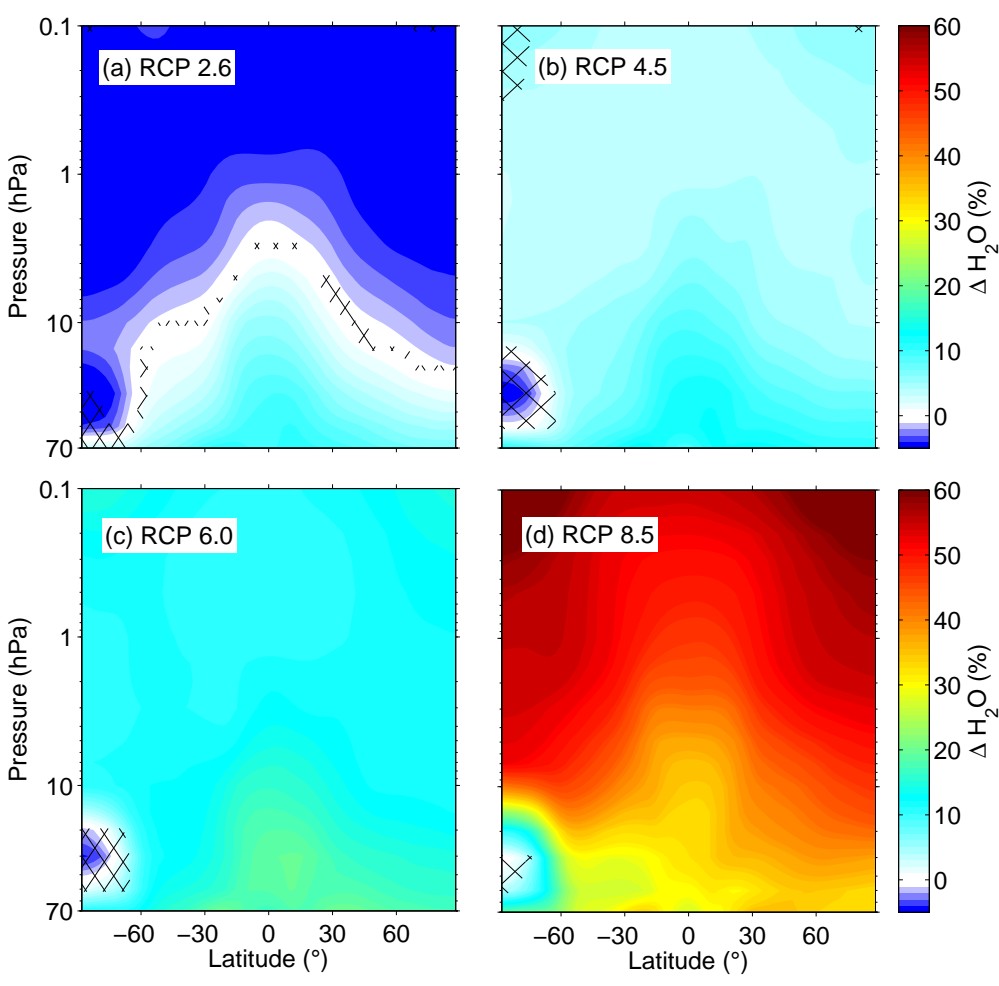

**Figure 10.** Change in water vapour between the 2000s and 2090s (2090s minus 2000s). Hatching indicates that the difference is not statistically significant at the 95% level of confidence, calculated with the Student's t-test.



**Table 1.** Summary of boundary conditions used for the SOCOL v.3 CCM simulations.

| Simulation | Period | Greenhouse gases | Ozone precursor emissions | Sea surface temperatures |
|---|---|---|---|---|
| RCP 2.6 | 2000-2100 | Observations until 2005 then RCP 2.6[a] | RCP 2.6 | RCP 2.6 |
| RCP 4.5 | 2000-2100 | Observations until 2005 then RCP 4.5[b] | RCP 4.5 | RCP 4.5 |
| RCP 6.0 | 1960-2100 | Observations until 2005 then RCP 6.0[c] | Historical emissions until 2000[d] then RCP 6.0 | CESM1(CAM5)[e] |
| RCP 6.0 -fEmis | 1960-2100 | Same as RCP 6.0 | Fixed at 1960 levels | Same as RCP 6.0 |
| RCP 6.0 -fEmis-fCH$_4$ | 1960-2100 | Same as RCP 6.0 for CO$_2$ and N$_2$O; CH$_4$ fixed at 1960 levels | Fixed at 1960 levels | Same as RCP 6.0 |
| RCP 8.5 | 2000-2100 | Observations until 2005 then RCP 8.5[f] | RCP 8.5 | RCP 8.5 |

[a] van Vuuren et al. (2011b)

[b] Thomson et al. (2011)

[c] Masui et al. (2011)

[d] Lamarque et al. (2010)

[e] Meehl et al. (2013)

[f] Riahi et al. (2011)