# Peer review of "The role of methane in projections of 21st century stratospheric water vapour"

_Atmospheric Chemistry and Physics, 2016_

## Referee Comment (RC1) · Anonymous Referee #1 · 15 Jul 2016

Further to my initial review I have the following specific comments.

1) There is occasional confusion (e.g. line 8 of the abstract) about the role of the changing BDC in this paper. Increasing BDC will of course creat more methane flux into the stratosphere but it will not necessarily cause an increase in SWV directly through increased advection because it cools the tropopause cold point. Please make this clear.

2) p2 Line 29 is also unclear about the role of the BDC increase. Is this not simply an increase in methane flux entering the stratosphere. Please reword this sentence for clarity.

3) There is a problem with the uncertainty envelopes in Figure 1. The authors show the interannual standard deviations if I understand correctly. However, this is not the

uncertainty on the multiannual means plotted in the solid lines. The envelopes should be narrowed by a factor equal to the square root of the number of years included to give the uncertainty in the means.

4) Line 26 argues that the too moist stratosphere can not be due to the tropopause cold point temperature bias as this is too cold in the model. However, as the authors state elsewhere, the SWV is dependent on the minimum temperature that air parcels experience on their transit - this is more related to the coldest regional temperatures at the tropopause, e.g. in the west pacific. So is the model cold bias everywhere? we need more than the zonal mean here.

5) p4 Line 33 argues that the results will not be biassed by the incorrect seasonal phasing of water vapour entry into the stratosphere. This may not be quite right as the BDC increases most in winter under climate change. The authors should at least acknowledge this.

6) p5 line 15: most or all CC models?

7) p8 Line 22: Joshi et al, GRL, 2006 were the first to point out circulation changes due to stratospheric water vapor.

8) p9 line 3: can the authors give any estimate of the relative size of surface climate warming due to stratospheric water vapor compared to say greenhouse gases?

9) Figure 4 panel c shows an apparently decreasing trend in cold point temperature variance. This could be important given the non-linearity of saturated water vapor pressure with temperature. Is it due to changing tropospheric variability in ENSO for example? Scaife et al 2003 and Garfinkel et al 2013 discuss the effects of tropospheric variability on SWV which ought to be considered here.
* * *

---

## Referee Comment (RC2) · Anonymous Referee #2 · 25 Jul 2016

In this manuscript authors use chemistry climate model (SOCOL) simulations to show that in future, the warmer tropical tropopause temperatures and stronger Brewer-Dobson circulation would lead to increased CH4 flux in the stratosphere. Therefore they suggest that increased stratospheric CH4 can lead to up significant enhancement in the upper stratospheric water vapour, but relatively smaller changes in the lower stratospheric water vapour. Overall, this is well written manuscript and can be published in ACP if authors can address following minor comments.

My main concern is the model's ability to simulate stratospheric water vapour correctly. Figures 1 and 2 show significant biases against observations. On Page 4 (line 32) you say that "Because the aim of this study is to analyse future changes in SWV on a multi-decadal scale, we do not anticipate that the model's shifted annual cycle in SWV will substantially bias our results"

[Figure]

But I don't think only wording is enough. As far as know most of the models have great difficulty in simulating lower stratospheric water vapour correctly as this is combination of TTL as well as stratospheric process. But if model is missing some key process, how can it simulate future water vapour changes correctly? Can you show some other analysis that can confirm that your model is good enough to study stratospheric water vapour changes, especially in the lower stratosphere? For e.g. is it possible to show show decrease in tropical lower stratospheric ozone/ TTL temperatures when BD circulation is stronger (even in supplementary material would be good enough)

Technical corrections Page 1

1) Line 2: Abstract: why only 21st century. I think this was and will always be true. Delete "21st century" 2) line 4: "circulation, so that more methane" 3) line 8: "Although, methane contribution of SWV maximise in the upper stratosphere, modelled" 4) line 16: repeated "ozone depletion"; just delete second one.

Page 2 5)Line 4: Oltmans and Hofmann (Nature, 1995) is better reference for Boulder data. 6)Line 5-10: Very long (and confusing) sentence so reword it. 7)line 10:- "merged" or various satellite data sets 8) Line 13: Explain that now you are discussing modelling studies. 9) Line 23: "that controls amount of water vapour entering in the stratosphere" 10) Line 32: Forster et al., (2007) discussed reduced ozone in the tropical lower stratosphere and colder CPT. Randel et al., and Dhomse et al, discussed enhanced upwelling/ adiabatic expansion of TTL causing colder temperatures.

Page 3: 11.line 6: First, we assess 12. line 17: sits???-> lies near 100 hPa 13: Figure 1: almost 25\% negative bias compared to MIPAS. 14. line 30: (Stenke et al., 2013)

page 6 line 7: Do you mean lower mesospheric temperatures? line 14:Chemistry Climate Model used in Dessler et al., 2013

Page 7 Line 10 and 13: Repeated "because" line 30: Compared to Page 8: Line 19: or "dehydration due to PSCs?"

---

## Author Comment (AC1) · 23 Sep 2016

Response to Reviewer 1
*Reviewer comments are shown in black, author responses in red, and changes to the text are in blue.*

1) There is occasional confusion (e.g. line 8 of the abstract) about the role of the changing BDC in this paper. Increasing BDC will of course create more methane flux into the stratosphere but it will not necessarily cause an increase in SWV directly through increased advection because it cools the tropopause cold point. Please make this clear.

We have clarified this throughout the paper. For example:
- Climate models project that increasing surface temperatures will increase the rate of tropical upwelling (Butchart and Scaife, 2001; Butchart et al., 2010). This means that more methane enters the stratosphere, which will further affect SWV (Austin et al., 2007). However, enhanced tropical upwelling will not necessarily lead to increases in SWV as adiabatic expansion of the tropical tropopause layer cools the CPT (Randel et al., 2006; Dhomse et al., 2008).
- Figure 4b shows that the rate of tropical upwelling is projected to increase through the 21st century, as simulated by other CCMs (Butchart et al., 2010; SPARC CCMVal-2, 2010). Stronger tropical upwelling transports more methane from the troposphere into the stratosphere where it undergoes subsequent oxidation to water vapour via Reaction R1 (Austin et al., 2007). Stronger tropical upwelling has also been shown to lead to reduced ozone concentrations in the tropical lower stratosphere (Bekki et al., 2013), resulting in cooling of the CPT and subsequent decreases in SWV (Randel et al., 2006; Dhomse et al., 2008).

2) p2 Line 29 is also unclear about the role of the BDC increase. Is this not simply an increase in methane flux entering the stratosphere. Please reword this sentence for clarity.

We have done so, as detailed in the first bullet point above.

3) There is a problem with the uncertainty envelopes in Figure 1. The authors show the interannual standard deviations if I understand correctly. However, this is not the uncertainty on the multiannual means plotted in the solid lines. The envelopes should be narrowed by a factor equal to the square root of the number of years included to give the uncertainty in the means.

As noted in the caption of Figure 1, we show the standard deviations rather than the standard error (standard deviation divided by the square root of the number of years). Since we want to show the range of potential uncertainties (as indicated by the standard deviations), we decided to keep the uncertainty envelopes as they are.

4) Line 26 argues that the too moist stratosphere can not be due to the tropopause cold point temperature bias as this is too cold in the model. However, as the authors state elsewhere, the SWV is dependent on the minimum temperature that air parcels experience on their transit - this is more related to the coldest regional temperatures at the tropopause, e.g. in the west pacific. So is the model cold bias everywhere? We need more than the zonal mean here.

We have added a panel to Figure 1 comparing modelled equatorial temperatures with radiosonde measurements (the same dataset that was used in Figure 1a) as a function of longitude. This new plot (below) shows that, although SOCOL agrees with measurements within one standard deviation, mean modelled temperatures are colder than observations and there is no warm bias in the model at the coldest point.

[Figure]

(d) Annual-mean modelled equatorial CPT temperature (90 hPa) compared with radiosonde measurements of the CPT as a function of longitude. The grey shaded areas represent one standard deviation either side of the observations and the dashed blue lines indicate one standard deviation either side of model data.

5) p4 Line 33 argues that the results will not be biased by the incorrect seasonal phasing of water vapour entry into the stratosphere. This may not be quite right as the BDC increases most in winter under climate change. The authors should at least acknowledge this.
We removed this sentence, but did also add that the BDC is strongest in winter in our discussion of Figure 1a:
Modelled temperatures in the tropical lower stratosphere reflect the behaviour of the Brewer-Dobson circulation well; a colder environment is simulated during the boreal cold period when the Brewer-Dobson circulation is stronger.

6) p5 line 15: most or all CC models?
All – we have changed this:
Figure 4b shows that the rate of tropical upwelling is projected to increase through the 21st century, as simulated by other CCMs (Butchart et al., 2010; SPARC CCMVal-2, 2010).

7) p8 Line 22: Joshi et al, GRL, 2006 were the first to point out circulation changes due to stratospheric water vapor.
Thank you – we have included this reference:
Such cooling through the 21st century is also expected to affect circulation patterns in the lower stratosphere and troposphere (Joshi et al., 2006; Maycock et al., 2013).

8) p9 line 3: can the authors give any estimate of the relative size of surface climate warming due to stratospheric water vapor compared to say greenhouse gases?
As we do not have a simulation with only stratospheric water vapour changing, we cannot say anything about the relative influence on surface climate.

9) Figure 4 panel c shows an apparently decreasing trend in cold point temperature variance. This could be important given the non-linearity of saturated water vapor pressure with temperature. Is it due to changing tropospheric variability in ENSO for example? Scaife et al 2003 and Garfinkel et al 2013 discuss the effects of tropospheric variability on SWV which ought to be considered here.

While we are interested in long-term trends in SWV rather than variability, we do know that there is an increased tendency towards ENSO conditions in the RCP 6.0 SSTs we prescribed for our simulations, so this may contribute to the upward trend in SWV. We have noted this:

Increases in the amount of water vapour entering the stratosphere are also linked with El Nino/Southern Oscillation conditions (Scaife et al., 2003), which occur more frequently throughout the 21st century under RCP 6.0.

---

## Author Comment (AC2) · 23 Sep 2016

Response to Reviewer 2
*Reviewer comments are shown in black, author responses in red, and changes to the text are in blue.*

In this manuscript authors use chemistry climate model (SOCOL) simulations to show that in future, the warmer tropical tropopause temperatures and stronger Brewer-Dobson circulation would lead to increased CH4 flux in the stratosphere. Therefore they suggest that increased stratospheric CH4 can lead to up significant enhancement in the upper stratospheric water vapour, but relatively smaller changes in the lower stratospheric water vapour. Overall, this is well written manuscript and can be published in ACP if authors can address following minor comments.

My main concern is the model's ability to simulate stratospheric water vapour correctly. Figures 1 and 2 show significant biases against observations. On Page 4 (line 32) you say that "Because the aim of this study is to analyse future changes in SWV on a multi-decadal scale, we do not anticipate that the model's shifted annual cycle in SWV will substantially bias our results" But I don't think only wording is enough. As far as know most of the models have great difficulty in simulating lower stratospheric water vapour correctly as this is combination of TTL as well as stratospheric process. But if model is missing some key process, how can it simulate future water vapour changes correctly? Can you show some other analysis that can confirm that your model is good enough to study stratospheric water vapour changes, especially in the lower stratosphere? For e.g. is it possible to show show decrease in tropical lower stratospheric ozone/ TTL temperatures when BD circulation is stronger (even in supplementary material would be good enough)

Our results show that the model has a significant bias in the simulated seasonal cycle of water vapour in the middle stratosphere. The annual mean properties (the main aim of the paper) are mostly well captured throughout the stratosphere. As shown in Figure 1a, temperature in the tropical lower stratosphere reflects the behavior of the Brewer-Dobson circulation well; a colder environment is simulated during the boreal cold period when the BDC is stronger. The same feature in the seasonal behavior of ozone is also well simulated, as shown by an additional panel in Figure 1:

[Figure]

(c) Climatological-mean modelled (4.2N-4.2S, 90 hPa) annual cycle in ozone compared with merged SWOOSH observations (3.75N-3.75S, 82.5 hPa) between 1984-2015.

Technical corrections
1) Line 2: Abstract: why only 21st century. I think this was and will always be true. Delete "21st century"
Changed as suggested:

Key processes driving changes in SWV include dehydration of air masses transiting the cold-point tropopause (CPT) and methane oxidation.

2) line 4: "circulation, so that more methane"
This sentence was deleted in the revised version of the manuscript.

3) line 8: "Although, methane contribution of SWV maximise in the upper stratosphere, modelled"
Changed:
Although the methane contribution to SWV maximises in the upper stratosphere, modelled SWV trends are found to be driven predominantly by warming of the CPT rather than by increasing methane oxidation.

4) line 16: repeated "ozone depletion"; just delete second one.
Changed:
Increases in SWV lead to stratospheric ozone depletion both by enhancing odd hydrogen cycles, and by increasing the prevalence of polar stratospheric clouds (PSCs) which facilitate polar springtime ozone depletion (Kirk-Davidoff et al., 1999; MacKenzie et al., 2004; Stenke and Grewe, 2005).

5)Line 4: Oltmans and Hofmann (Nature, 1995) is better reference for Boulder data.
We have included this reference:
Between 1980 and 2010, balloon-borne measurements of SWV over Boulder, Colorado (40N) (Oltmans and Hofmann, 1995) showed an increase in SWV of ~1 ppmv (almost 30%) between 16-26 km, with ~0.25 ppmv of the net increase due to increased methane oxidation in the stratosphere (Hurst et al., 2011).

6)Line 5-10: Very long (and confusing) sentence so reword it.
Changed:
This inferred contribution of methane oxidation to the SWV trend is in good agreement with an earlier analysis (Rohs et al., 2006) of balloon-borne SWV measurements from Japan, France and Sweden, with launch sites were between 39-68N. Rohs et al. (2006) concluded that methane contributed 25-34% to the increase in water vapour in the middle stratosphere between 1978 and 2003, and a smaller amount (1.6-10.7%) in the lower stratosphere. The remaining increase was attributed to increased transport of water vapour from the troposphere to the stratosphere.

7)line 10:- "merged" or various satellite data sets
Changed:
More recently, Hegglin et al. (2014) showed that merged SWV observations from various satellite datasets display positive trends globally in the upper stratosphere due to methane increases and changes in stratospheric circulation, but negative trends in the lower stratosphere.

8) Line 13: Explain that now you are discussing modelling studies.
Unchanged:
Austin et al. (2007) also investigated the causes of past changes in SWV, but used a coupled chemistry-climate model (CCM) to do so.

9) Line 23: "that controls amount of water vapour entering in the stratosphere"
Changed:

Transport and evaporation of ice through the tropical tropopause is also a significant process that controls the amount of water vapour entering the stratosphere (Dessler et al., 2016).

10) Line 32: Forster et al., (2007) discussed reduced ozone in the tropical lower stratosphere and colder CPT. Randel et al., and Dhomse et al, discussed enhanced upwelling/ adiabatic expansion of TTL causing colder temperatures.
We have clarified this:
Climate models project that increasing surface temperatures will increase the rate of tropical upwelling (Butchart and Scaife, 2001; Butchart et al., 2010). This means that more methane enters the stratosphere, which will further affect SWV (Austin et al., 2007). However, enhanced tropical upwelling will not necessarily lead to increases in SWV as adiabatic expansion of the tropical tropopause layer cools the CPT (Randel et al., 2006; Dhomse et al., 2008).

Page 4:
11.line 6: First, we assess
Changed:
First, we assess the fidelity of SOCOL v.3's simulation of water vapour, temperature and methane through comparison with observations.

12. line 17: sits???-> lies near 100 hPa
Changed:
In SOCOL v.3 the WMO-defined tropopause (the lowest level at which the lapse rate decreases to 2 °C km$^{-1}$ or less) typically lies near 100 hPa in the tropics, while the CPT typically locates at a lower pressure of 90 hPa.

13: Figure 1: almost 25% negative bias compared to MIPAS.
We have noted this in the revised version:
However, the modelled annual cycle is shifted such that the annual maximum occurs a month earlier, in September rather than October (discussed further below), and SOCOL exhibits an almost 25% negative bias compared with observations between January - April.

14. line 30: (Stenke et al., 2013)
Changed:
The annual cycle in water vapour progresses with decreasing pressure (Randel et al., 1998), and because SOCOL v.3 has too-fast upward propagation as estimated from the water vapour tape recorder (Stenke et al., 2013), seasonal variability is shifted upwards such that maximum water vapour mixing ratios occur out-of-phase with observations.

line 7: Do you mean lower mesospheric temperatures?
Yes – we have corrected this:
Between 1960 and 2100, stratospheric and mesospheric temperatures are projected to decrease, with a maximum cooling of 12 K projected for the mesosphere in the RCP 6.0-fEmis-fCH4 simulation (Fig. 5c).

line 14:Chemistry Climate Model used in Dessler et al., 2013
Corrected:
This is akin to looking at the change in SWV in our RCP 6.0-fEmis-fCH4 simulation (Fig. 5a) and indeed the results are very similar, with both the GEOSCCM (Goddard Earth Observing System Chemistry

Climate Model used by Dessler et al., 2013) and SOCOL v.3 models simulating an increase in SWV due to climate-related changes of ~1 ppmv through the 21st century.

Line 10 and 13: Repeated "because"
This was intentional.

line 30: Compared to Page 8: Line 19: or "dehydration due to PSCs?"
Sorry, this comment is not clear to us – perhaps the wrong page and line numbers have been referenced.